Effects of mannan oligosaccharides on growth performance, nutrient digestibility, ruminal fermentation and hematological parameters in sheep

Zheng Chen 1
Zhou Juwang 1
Zeng Yanqin 1 2
Liu Ting 1 liuting@gsau.edu.cn
1 Gansu Agricultural University , Lanzhou , China 中国
2 Lanzhou University , Lanzhou , China 中国
Okpala Charles
Electronic publication date: 2021 Jun 30
Publication date: 2021
Volume: 9
Electronic Location ID: e11631
Received 2021 Jan 9; Accepted 2021 May 27
Copyright: © 2021 Zheng et al.
Copyright year: 2021
Copyright holder: Zheng et al.
License: This is an open access article distributed under the terms of the Creative Commons Attribution License, which permits unrestricted use, distribution, reproduction and adaptation in any medium and for any purpose provided that it is properly attributed. For attribution, the original author(s), title, publication source (PeerJ) and either DOI or URL of the article must be cited.
License URL: https://creativecommons.org/licenses/by/4.0/

Keywords: Digestibility, Growth performance, Mannan oligosaccharides, Ruminal fermentation, Sheep

Funding: National Natural Science Foundation of China 31860657 This study was supported by the National Natural Science Foundation of China (No. 31860657). The funders had no role in study design, data collection and analysis, decision to publish, or preparation of the manuscript.

==============================
Background

Mannan oligosaccharides (MOS) are a promising feed additive in animal husbandry due to mainly improving animal health status. The purpose of this study was to investigate the effects of MOS on growth performance, nutrient digestibility, ruminal fermentation, and twelve hematological parameters in sheep.

Methods

Ninety-six healthy Hu rams with similar body weights were chosen and divided into four treatment groups (twenty-four rams in each group), in which four different doses of MOS were tested: 0%, 0.8%, 1.6% and 2.4% of the basal diet (on an as-fed basis).

Results

The results showed that supplementation dietary MOS did not affect feed intake, body weight, average daily weight gain, or ruminal short-chain fatty acids (SCFAs) concentration; the ratio of individual fatty acids to total SCFAs, the C2/C3 ratio, and the hematological parameters in the sheep were also unaltered (P > 0.05). Conversely, supplementation dietary MOS increased the dry matter, organic matter, crude protein, neutral detergent fiber, acid detergent fiber, and ash apparent digestibility (P < 0.05), and decreased the ruminal ammonia concentration in the sheep (P < 0.05), especially at a dose of 1.6%.

Conclusions

This indicates that supplementation dietary MOS improved nutrient utilization by the sheep and nitrogen metabolism in the rumen; however, the effects are too slight to interfere with the basal metabolism in the sheep.

Introduction

In the sheep industry, the sheep should be fed by balanced diets which contain all kinds of nutrients even some small feed additives to improve nutrients digestion and absorption, promote ruminal fermentation and hematological parameters, ultimately enhance the growth performance and productivity of sheep. It is well known that more nutrients are digested and absorbed, better ruminal fermentation is established, and healthier body status including hematological parameters is formed, will lead to higher growth performance and productivity in sheep.

The use of functional feed ingredients, such as probiotics, prebiotics, and immunostimulants, to improve species growth and product quality without harming the environment has recently become widespread. Prebiotics are good examples of food supplements that improve animals’ growth performance and benefit animals’ health by modulating gastrointestinal tract microbiota, such as providing energy for favorable endogenous bacteria and reducing enumeration of pathogenic intestinal bacteria (Abd El-Hack et al., 2021).

Mannan oligosaccharides (MOS) are structural cell wall components of Saccharomyces cerevisiae (Van den Abbeele et al., 2020). In monogastric animals, supplementation dietary MOS brought uplifting effects including improvement growth performance and nutrients digestion, promotion health status, boosting gastrointestinal tract morphological integrity, enhancement antioxidant capacity, facilitation immunological status, upregulation immune related gene expression (Ren et al., 2020; Van den Abbeele et al., 2020; Zhou et al., 2020).

Because of special physiological feature and gastrointestinal tract structure, fewer researches focused on the effects of MOS on ruminants due to a principle widely accepted that ruminal microbe degraded oligosaccharides and weakened their activity (Zheng et al., 2018). In fact, however, supplementation dietary MOS has been confirmed that they benefited their ruminal hosts. In beef cattle and dairy cows, supplementation dietary MOS maintained more body weight of beef cows during parturition (Linneen et al., 2014), produced significantly more colostrum of dairy cows (Westland et al., 2017), enhanced immune response to rotavirus of dairy cows during the dry period and tended to enhance the subsequent transfer of rotavirus antibodies to their calves (Franklin et al., 2005). In sheep, supplementation dietary MOS increased ruminal pH and total short chain fatty acids (SCFAs) concentration, decreased lipopolysaccharides (LPS) level in plasma, reduced ruminal ammonia concentration and ruminal stratum corneum thickness and total thickness of ruminal epithelium, as well as the incidence and severity of hepatic abscesses (Diaz et al., 2018). In our previous studies, similarly, supplementation dietary MOS increased antioxidant capacity (Zheng et al., 2018) and crude protein (CP) retention rate and decreased energy release as methane (CH4) in sheep (Zheng et al., 2019).

Nevertheless, many previous studies, including our previous studies, have been carried out under experimental feeding conditions in laboratories; thus, the actual breeding effects of MOS on sheep under practical production conditions still need be revealed. In addition, based on these aforementioned achievements, such as supplementation dietary MOS improved CP and energy utilization in sheep and maintained well healthy and production status in cattle under laboratories conditions, we hypothesized that supplementation dietary MOS could improve productivity and nutrient utilization in sheep under actual farm condition, and sought to determine changes in several hematological parameters in sheep because of insufficient evidence provided by hematological parameters. Therefore, this study examined growth performance, apparent nutrient digestibility, ruminal fermentation, and some hematological parameters in sheep fed MOS to provide evidences supporting the actual breeding application of MOS in ruminants’ feed.

Materials & methods

All experiments in this study were carried out in accordance with the approved guidelines of the Regulation Standing Committee of Gansu People’s Congress. All experimental protocols and sample collection were approved by the Ethics Committee of Gansu Agriculture University under permission no. GAU-LC-2020-018.

Schematic overview of the experimental program

The experiments included sheep feeding trial and samples laboratory analysis. The sheep were fed for 7 weeks, the feed intake and growth performance were recorded, and the feces, urine, ruminal fluid, and blood samples were collected during the feeding trial. The blood samples were analyzed by a fully automatic hematology analyzer immediately after collecting. Then, the other samples were taken back to laboratory to analyze the nutrients concentration and ruminal fermentation parameters (Fig. 1).

Figure 1 Schematic overview of the experimental program.

Experimental design, animals, and housing

A single-factor design was used for this experiment. Four different doses of MOS (SCIPHAR®; Sciphar, Inc., Xi’an, Shaanxi, P. R. China) were tested: 0%, 0.8%, 1.6% and 2.4% of basal diet (on an as-fed basis). Ninety-six healthy Hu rams with similar body weights (31.11 ± 2.69 kg) were chosen and divided into four treatment groups. Each group had twenty-four rams. The test period included a 2-week acclimation period and a 5-week feeding trial (including a 6-day digestion and metabolism experiment during the fourth week). All rams were loose-housed in 10 m2 barns and 20 m2 yards according to experimental design. Each loose-housing system was equipped with a feeder and a drinker, providing ad libitum access to feed and water.

During the feeding trial, the weights of feed intake and residues were carefully recorded daily in each replicate, and body weights were recorded weekly before the morning feeding.

Experimental diets

Four isoenergetic and isonitrogenous diets were formulated to meet or exceed nutrient recommendations for rams according to the China Agricultural Industry Standard (NY/T816-2004). Feedstuff’s compositions were adjusted to create four experimental diets containing different levels of MOS but the same nutrient compositions (Table 1).

Table 1 Ingredients and chemical composition of experimental diets.

Items	MOS2	
0%	0.8%	1.6%	2.4%	
Ingredients, g/kg					
Maize	400.0	400.0	427.4	456.5	
Soybean meal	120.0	130.0	135.0	150.0	
Sunflower skin	90.0	90.0	89.0	90.0	
Malt sprout	122.2	104.2	108.8	85.7	
Barley	63.0	67.0	30.0	30.0	
Cottonseed meal	5.0	5.0	2.0	3.0	
Maize straw	160.0	156.0	152.0	148.0	
Salt	6.8	6.8	6.8	6.8	
Puffing urea	13.0	13.0	13.0	13.0	
Sodium bicarbonate	15.0	15.0	15.0	15.0	
Mineral and vitamin mix1	5.0	5.0	5.0	5.0	
Mannan oligosaccharides (MOS)	–	8.0	16.0	24.0	
Chemical composition, g/kg DM					
Crude protein	182.6	182.4	181.8	181.8	
Calcium	3.9	3.8	3.8	3.6	
Phosphorus	3.2	3.1	3.1	3.0	
Neutral detergent fiber (NDF)	317.3	309.0	299.5	285.2	
Acid detergent fiber (ADF)	193.6	189.9	185.6	181.0	
Starch	333.7	333.8	331.5	333.4	
Ether extract	21.3	20.9	21.0	20.6	
Ash	32.1	32.9	31.2	30.8	
Metabolic energy, MJ/kg DM	10.4	10.4	10.3	10.3	
Notes:

1 Mineral and vitamin mix was formulated with 200 mg/kg of S, 25.0 mg/kg of Fe, 40.0 mg/kg of Zn, 8 mg/kg of Cu, 0.3 mg/kg of I, 40.0 mg/kg of Mn, 0.2 mg/kg of Se, 0.1 mg/kg of Co, 940 IU/kg of vitamin A, and 20 IU/kg of vitamin E.

2 MOS, mannan oligosaccharides.

Sample collection and analysis

Feed, feces and urine samples collection and preparation

During the third and fourth weeks, the digestion and metabolism experiment was carried out (Costa et al., 2021; Sinz et al., 2021). In brief, six sheep was randomly chosen from each treatment, and the 24 chosen sheep were transferred into individual digestion and metabolism cages. After a 6-day acclimation period, 10% of the total feed, 10% of the total feces output, and 5% of the total urine output (5 mL sulfuric acid was added to the collection bowl to prevent nitrogen release before collection) were sampled daily for 6 days and stored at −20 °C (Zheng et al., 2018). For nitrogen analysis, samples of 3% of the total feces output were taken daily, stored in wide-mouth bottles with 20 mL 10% sulfuric acid, and pooled after 6 days (Zheng et al., 2018). At the end of the data collection period, feed and fecal samples were thawed and pooled for each sheep, and then dried at 65 °C for 72 h in a forced-air oven for partial dry matter (DM) determination. Dried, pooled feed and fecal samples were ground through a 1 mm screen in a Wiley mill (Ogawa Seiki Co., Ltd., Tokyo, Japan).

Ruminal fluid and blood samples collection

After the digestion and metabolism experiment, ruminal fluid and blood samples were collected 3 h after feeding. Approximately 50 mL of ruminal fluid was taken from three regions of the rumen through the mouth of each sheep using a flexible pipe and syringe, strained through four layers of cheesecloth, and preserved in individual plastic tubes (Jolazadeh et al., 2019). Then, a 5 mL blood sample was collected from the jugular vein into a non-heparinized vacuum tube from each sheep.

Nutrients analyses in feed, feces and urine samples

The nutrients concentration of feed, feces, and urine samples were examined according to the following Association of Official Analytical Chemists methods (AOAC, 2002): analytical DM (method 930.15), ash (method 942.05), calcium (Ca, method 978.02), total phosphorus (TP, method 946.06); neutral detergent fiber (NDF) and acid detergent fiber (ADF) were analyzed according to the methods of Goering & Soest (1970). The nitrogen concentration of feed, feces with 10% sulfuric acid, and urine samples were determined by the Kjeldahl method (AOAC, 2002, method 990.03).

Short-chain fatty acids analysis in ruminal fluid

Short-chain fatty acids (SCFAs) were measured using an Agilent 6890N gas chromatography system (Agilent Technologies, Inc., Santa Clara, CA, USA) with a 30 m (0.32 mm i.d.) fused silica column (HP-19091N-213I; Agilent, Santa Clara, CA, USA) (Zheng et al., 2018). In brief, the ruminal fluid sample was centrifuged at 5,400 rpm for 10 min, and then 1 mL supernatant was transferred into a 1.5 mL centrifugal tube. Then 200 μL 25% metaphosphoric acid solution contained 2 g/L 2-ethylbutyric acid was added into tube and the tube was put into ice-water for more than 30 min. After that, the tube was centrifuged at 10,000 rpm for 10 min, and the supernatant was analyzed by gas chromatography. The gas chromatography condition was as follow: injection port temperature 220 °C; sample size 0.6 μL; carrier gas nitrogen; nitrogen flow rate 2 mL/min; split ratio 40:1; oven temperature 120 °C for 3 min, then 10 °C/min rising until 180 °C, maintaining for 1 min; flame ionization detector temperature 250 °C, hydrogen flow rate 40 mL/min, air flow rate 450 mL/min, make-up gas flow rate 45 mL/min.

Ammonia nitrogen concentration analysis in ruminal fluid

The ammonia nitrogen concentration was measured using a spectrophotometer (SP-723; Spectrum Instruments, Ltd., Shanghai, P. R. China) according to the Berthelot reaction (phenol-hypochlorite) described by Broderick & Kang (1980). In brief, the ruminal fluid sample was centrifuged at 4,000 rpm for 10 min, and then 2 mL supernatant was transferred into a 15 mL tube. Then 8 mL 0.2 mol/L hydrochloric acid solution was added into the tube. 0.4 mL solution was transferred into a tube and 2 mL solution A (0.08 g sodium nitroferricyanide dihydrate dissolved in 14% sodium salicylate solution) and 2 mL solution B (2 mL sodium hypochlorite dissolved in 100 mL 0.3 mol/L sodium hydroxide solution) were added. The solution was shaken up, and placed in tube shelf for reaction for 10 min. After that, the ammonia concentration was measured by colorimetry at 700 nm wave length to obtain optical density. While optical density from different concentration ammonia standard solution also needed measurement to calculate the ammonia concentration in ruminal fluid.

Hematological parameters analysis in blood

After blood samples collection, parameters including white blood cell (WBC) and red blood cell (RBC) counts, hemoglobin (HGB) concentration, hematocrit (HCT), mean corpuscular volume (MCV), mean corpuscular hemoglobin (MCH), mean corpuscular hemoglobin concentration (MCHC), red blood cell distribution width (RDW), platelet count (PLT), mean platelet volume (MPV), platelet distribution width (PDW), and plateletcrit (PCT) were measured immediately using a fully automatic hematology analyzer (Mindray BC-2800Vet, Shenzhen Mindray Bio-Medical Electronics Co. Ltd., Shenzhen, Guangdong, P. R. China).

Statistical analysis

Because the single-factor experimental design was used in this experiment, the data were analyzed by one-way ANOVA (SPSS 19.0; IBM Co., Chicago, IL, USA) using the following model:

Xij = μ + αi + eij

where Xij is the observation of the dependent variable (i = 1 to 4, j = 1 to 6), μ is the population mean, αi is the ith treatment effect, and eij is the random error associated with the observation.

Significance was declared at P ≤ 0.05 and tendency at 0.05 < P ≤ 0.10 using Tukey’s multiple comparison test.

Results

Effects of different doses MOS on sheep growth performance

There were no differences in final body weight, average daily weight gain, or average daily feed intake between treatment groups (P > 0.05, Table 2).

Table 2 Effects of different doses of MOS on growth performance in sheep.

Items1	MOS2	SEM3	P-value	
0%	0.8%	1.6%	2.4%	
Initial weight, kg	30.82	31.14	31.25	31.25	0.276	0.940	
Average daily intake, g/d	1595	1540	1556	1481	27.56	0.551	
Final weight, kg	38.37	37.74	38.77	38.58	0.374	0.782	
Average daily gain, g/d	207.1	190.9	213.5	214.6	7.578	0.667	
Notes:

1 Sheep were fed 0, 0.8, 1.6, or 2.4% MOS (n = 24 per treatment). The mean growth performance results for the 96 sheep according to MOS level are shown for the 5-week collection phase of the study.

2 MOS, mannan oligosaccharides.

3 SEM, standard error of the mean.

Effects of different doses MOS on nutrient apparent digestibility and retention rate

The DM and OM apparent digestibility of sheep fed the 1.6% MOS diet were higher than those in sheep fed the 0% MOS diet (P = 0.010, P = 0.016, Table 3), and their NDF and ADF apparent digestibility were higher than those in the other treatment groups (P = 0.003, P = 0.028, P = 0.001; P = 0.001, P = 0.029, P < 0.001). In addition, the CP apparent digestibility of sheep fed the 1.6% MOS diet was higher than those in sheep fed the 0% and 0.8% MOS diet (P = 0.040, P = 0.044, Table 4). There were no differences in the apparent digestibility of other nutrients or in the retention rates of sheep between treatment groups (P > 0.05).

Table 3 Effects of different doses of MOS on apparent digestibility of DM, OM, Ash, NDF and ADF in sheep.

Items1	MOS2	SEM3	P-value	
0%	0.8%	1.6%	2.4%	
DM4							
Intake, g/d	1118	1226	1103	1271	58.72	0.717	
DM in feces, g/d	367.9	375.8	313.1	382.7	17.52	0.508	
Digested DM, g/d	750.0	849.7	789.7	888.6	42.76	0.696	
Apparent digestibility, %	66.66b	68.93ab	72.01a	69.76ab	0.636	0.017	
OM5							
Intake, g/d	1021	1120	1014	1174	53.99	0.693	
OM in feces, g/d	303.9	316.4	262.4	315.5	15.08	0.573	
Digested OM, g/d	716.9	804.1	751.4	858.6	40.27	0.648	
Apparent digestibility, %	69.91b	71.40ab	74.70a	73.01ab	0.601	0.020	
Ash							
Intake, g/d	97.14	105.0	89.09	97.20	4.854	0.744	
Ash in feces, g/d	63.95	59.41	50.76	67.19	2.637	0.134	
Digested ash, g/d	33.19	45.61	38.33	30.01	2.980	0.281	
Apparent digestibility, %	32.54ab	42.60a	41.46ab	30.45b	1.744	0.014	
NDF6							
Intake, g/d	323.0	347.0	328.8	329.8	16.51	0.967	
NDF in feces, g/d	189.0	192.3	161.5	200.2	9.756	0.554	
Digested NDF, g/d	134.0	154.7	167.2	129.5	8.611	0.386	
Apparent digestibility, %	41.17b	43.87b	52.12a	39.14b	1.356	0.001	
ADF7							
Intake, g/d	156.8	170.0	161.1	166.6	8.109	0.951	
ADF in feces, g/d	110.7	110.6	93.71	118.7	5.648	0.484	
Digested ADF, g/d	46.13	59.35	67.37	47.96	3.718	0.138	
Apparent digestibility, %	29.06b	34.22b	42.95a	28.60b	1.529	<0.001	
Notes:

1 Sheep were fed 0, 0.8, 1.6, or 2.4% MOS (n = 6 per treatment). The mean digestion results are shown for each treatment over the 6-day collection phase of the study.

2 MOS, mannan oligosaccharides.

3 SEM, standard error of the mean.

4 DM, dry matter.

5 OM, organic matter.

6 NDF, neutral detergent fiber.

7 ADF, acid detergent fiber.

Means within rows with different superscript letters significantly differ (P < 0.05).

Table 4 Effects of different doses of MOS on retention rate of CP and apparent digestibility of CP, Ca, and TP in sheep.

Items1	MOS2	SEM3	P-value	
0%	0.8%	1.6%	2.4%	
CP4							
Intake, g/d	200.5	221.3	224.9	227.0	9.105	0.738	
CP in feces, g/d	63.34	69.74	63.19	64.16	2.483	0.777	
CP in urine, g/d	79.68	73.26	87.11	90.76	4.525	0.545	
Digested CP, g/d	137.1	151.6	161.7	162.8	7.083	0.571	
Apparent digestibility, %	67.90b	68.00b	72.16a	71.61ab	0.719	0.045	
Retained CP, g/d	57.45	78.33	74.61	72.06	6.134	0.662	
Retention rate, %	27.81	34.35	33.09	31.70	1.938	0.677	
Ca5							
Intake, g/d	5.470	5.578	5.232	6.260	0.282	0.637	
Ca in feces, g/d	5.342	5.365	4.811	5.818	0.214	0.451	
Digested Ca, g/d	0.127	0.213	0.422	0.442	0.133	0.816	
Apparent digestibility, %	−0.971	2.548	4.390	6.423	2.670	0.814	
TP6							
Intake, g/d	4.430	4.566	4.057	4.933	0.225	0.613	
TP in feces, g/d	3.600	3.371	3.186	3.544	0.133	0.716	
Digested TP, g/d	0.831	1.195	0.871	1.389	0.137	0.441	
Apparent digestibility, %	17.00	24.46	15.79	27.71	3.006	0.454	
Notes:

1 Sheep were fed 0, 0.8, 1.6, or 2.4% MOS (n = 6 per treatment). The mean of digestion and retention results are shown for each treatment over the 6-day collection phase of the study.

2 MOS, mannan oligosaccharides.

3 SEM, standard error of the mean.

4 CP, crude protein.

5 Ca, calcium.

6 TP, total phosphorus.

Means within rows with different superscript letters significantly differ (P < 0.05).

Effects of different doses MOS on ruminal fluid and hematological parameters

Supplementation dietary MOS did not affect SCFAs concentration, ratios of individual fatty acids to total SCFAs, or C2/C3 ratios (P > 0.05); however, the ammonia concentration (NH3-N) of ruminal fluid from sheep fed the 1.6% and 2.4% MOS diets were lower than that of sheep fed the 0% MOS diet (P = 0.035, P = 0.013, Table 5).

Table 5 Effects of different doses of MOS on ruminal fermentation parameters in sheep.

Items1	MOS2	SEM3	P-value	
0%	0.8%	1.6%	2.4%	
SCFAs4							
Total, mM/L	50.25	38.59	42.94	43.44	3.465	0.721	
Acetate, % of total	62.12	65.17	63.18	63.91	0.520	0.210	
Propionate, % of total	17.60	16.06	16.33	16.58	0.445	0.659	
Butyrate, % of total	13.49	11.96	14.38	13.51	0.687	0.683	
Iso-butyrate, % of total	1.923	2.200	1.870	1.859	0.137	0.816	
Valerate, % of total	1.318	1.189	1.139	1.152	0.066	0.789	
Iso-valerate, % of total	3.547	3.427	3.099	2.986	0.226	0.816	
C2/C35	3.595	4.072	3.911	3.997	0.121	0.550	
Ammonia (NH3-N), mg/100 mL	28.64a	20.78ab	19.79b	18.39b	1.286	0.011	
Notes:

1 Sheep were fed 0, 0.8, 1.6, or 2.4% MOS (n = 6 per treatment). The mean of ruminal fermentation parameters results are shown for each treatment after collection on the seventh day of the study.

2 MOS, mannan oligosaccharides.

3 SEM, standard error of the mean.

4 SCFAs, short-chain fatty acids.

5 C2/C3, acetate/propionate.

Means within rows with different superscript letters significantly differ (P < 0.05).

Similarly, supplementation dietary MOS did not affect the hematological parameters of sheep (P > 0.05), and there was only a tendency regarding MOS increasing the MCHC concentration in blood (P = 0.068, Table 6).

Table 6 Effects of different doses of MOS on hematological parameters in sheep.

Items1	MOS2	SEM3	P-value	
0%	0.8%	1.6%	2.4%	
WBC4, ×109/L	190.4	206.4	189.4	183.4	7.136	0.730	
RBC5, ×1012/L	13.23	14.17	13.58	13.18	0.233	0.432	
HGB6, g/L	137.0	142.3	136.5	138.6	2.482	0.849	
HCT7, %	45.60	46.23	44.12	43.58	0.791	0.644	
MCV8, fL	34.58	32.68	32.57	33.10	0.380	0.200	
MCH9, pg	10.32	10.00	10.00	10.46	0.104	0.321	
MCHC10, g/L	299.8	307.5	309.0	317.4	2.339	0.068	
RDW11, %	15.00	15.45	15.28	15.14	0.168	0.820	
PLT12, ×109/L	803.0	1074	902.5	749.0	65.68	0.337	
MPV13, fL	4.883	4.567	4.733	4.740	0.060	0.319	
PDW14	16.10	15.82	15.83	16.02	0.057	0.202	
PCT15, %	0.391	0.430	0.423	0.351	0.025	0.713	
Notes:

1 Sheep were fed 0, 0.8, 1.6, or 2.4% MOS (n = 6 per treatment). The mean of hematological parameter results is shown for each treatment after collection on the seventh day of the study.

2 MOS, mannan oligosaccharides.

3 SEM, standard error of the mean.

4 WBC, white blood cell.

5 RBC, red blood cell.

6 HGB, hemoglobin concentration.

7 HCT, hematocrit.

8 MCV, mean corpuscular volume.

9 MCH, mean corpuscular hemoglobin.

10 MCHC, mean corpuscular hemoglobin concentration.

11 RDW, red blood cell distribution width.

12 PLT, platelet counts.

13 MPV, mean platelet volume.

14 PDW, platelet distribution width.

15 PCT, plateletcrit.

Discussion

Effects of different doses MOS on sheep growth performance

Many studies have concluded that supplementation dietary MOS improve the performance of monogastric animals. Supplementation dietary MOS improved growth performance and gut health in broiler chickens (Soumeh et al., 2019), and increased feed utilization and amino acid digestibility in White Pekin ducks (Park, Jung & Carey, 2019), as well as improved growth performance and modulation of intestinal microbial populations in Japanese quail (Hazrati, Rezaeipour & Asadzadeh, 2019). Supplementation dietary MOS maintained the normal intestinal health of rats (Yazbeck et al., 2019), and effectively replaced antibiotics as growth promoters in raising guinea pigs (Minguez, Ingresa-Capaccioni & Calvo, 2019). Additionally, supplementation dietary MOS improved the growth performance, antioxidant capacity and immunity of aquatic animals (Lu et al., 2019; Meng et al., 2019; Mohammadian et al., 2019; Widanarni et al., 2019). However, the studies focused on MOS in ruminants have drawn inconsistent conclusions. Westland et al. (2017) reported that supplementation dietary MOS increased the colostrum yield of dairy cows, but had no effect on calf health and weight gain. Da Silva, Bittar & Ferreira (2012) observed that supplementation dietary MOS did not improve calf performance when added to a milk replacer or starter concentrate. Morrison, Dawson & Carson (2010) reported no effects on calf performance, even though supplementation with MOS increased concentrate intake in the early lives of calves. Similarly, supplementation dietary MOS have been shown not to influence body weight gain in lambs (Demirel et al., 2007). Consistently with some of these previous studies, supplementation dietary MOS did not impact average feed intake or average daily weight gain of sheep in the current study (refer to Table 2). This may be because ruminal microbes degrade MOS in the rumen (Dai et al., 2015; Wang et al., 2019), and then supplementation dietary MOS could regulate the composition of ruminal microbiota and chyme, even changed ruminal fermentation type and nutrients transfer in small intestine (Diaz et al., 2018) but the efficiency was not enough to change growth performance. This indicates that MOS are suitable for nutritional regulation purposes in adult ruminants, but not for growth performance enhancement in sheep.

Effects of different doses MOS on nutrient apparent digestibility and retention rate

In this study, the DM, OM, NDF, ADF, CP, and ash apparent digestibility of sheep were significantly increased through supplementation with MOS, especially at a dose of 1.6% (refer to Tables 3 and 4). MOS are a type of carbohydrate that can be degraded by ruminal microbes (Dai et al., 2015; Wang et al., 2019). Indeed, supplementation dietary MOS has potential to maintain stable ruminal environment by regulating ruminal microbiota, then the composition of chyme from rumen and to duodenum changed, consequently, some oligosaccharides went to small intestine because of MOS not degraded 100% in rumen and nutrients digestion and absorption in small intestine regulated by MOS (Diaz et al., 2018). Because supplementation dietary MOS brought long villi and shallow crypts which provided a larger surface area for nutrients absorption in small intestine of animals (Chacher et al., 2017), and an our previous study also demonstrated that supplementation dietary MOS elongated the villus height and the muscular thickness, and decreased the villus width of lamb duodenum significantly (Zheng et al., 2020), so, the monitoring results about supplementation dietary MOS improved nutrient digestion of sheep in the current study were observed. And some previous studies have reported similar results, with MOS increasing the digestion of NDF and ADF (Zheng et al., 2018), and nitrogen retention in sheep (Zheng et al., 2018; Zheng et al., 2019), as well as improving the nitrogen, zinc, and iron metabolism of calves and lambs (Cole, Purdy & Hutcheson, 1992). However, another study reported that MOS did not affect nutrient digestibility in dairy cows (Moallem et al., 2009). Thus, future research confirming the effects of MOS on nutrient utilization by ruminants is warranted.

Effects of different doses MOS on ruminal fluid and hematological parameters

In the current study, MOS did not change the SCFAs concentration or ratios of individual fatty acids to total SCFAs; however, it significantly decreased the ruminal ammonia concentration (refer to Table 5). In the rumen, lower ammonia level indicates active microbial proliferation, and microbial protein synthesis and nitrogen metabolism are improved (Diaz et al., 2018). As a result, nitrogen utilization by the sheep improved, which is consistent with our observations of nutrient utilization. Previous reports have indicated that supplementation dietary MOS improve the ruminal microbial composition and ammonia levels: the supplementation of a high-grain diet with MOS decreased ruminal ammonia concentration in sheep (Diaz et al., 2018) and slightly reduced ruminal ammonia concentration in sheep rumens 1–5 h after feeding (Zheng et al., 2018); live yeast supplementation also reduced ruminal ammonia concentration in dairy cows (Moallem et al., 2009). Other oligosaccharides have shown similar properties: the addition of chitosan to a 50:50 concentrate:forage diet decreased ruminal ammonia concentration in sheep (Goiri, Oregui & Garcia-Rodriguez, 2010), and the supplementation with β 1-4 galacto-oligosaccharides also reduced ruminal ammonia concentration in sheep (Mwenya et al., 2004). In addition, although ruminal SCFAs concentration was not influenced by MOS significantly, but supplementation dietary MOS decreased SCFAs concentration. This may be supplementation dietary MOS could increase SCFAs concentration (Diaz et al., 2018), and then more SCFAs could stimulate the expression of transport proteins like monocarboxylate transporter 1 (MCT1) and Na+/H+ exchangers (NHEs) and free fatty acid receptors (FFARs), consequently, more SCFAs were absorbed by ruminal epithelium (Baaske et al., 2020). As a result, the decrease of SCFAs concentration in sheep rumen was monitored in the current study because the ruminal fluid was sampled 3 h after feeding. These results illuminate the potential of MOS to improve ruminal fermentation and nitrogen utilization in ruminants, but the further research is guaranteed to confirm the actual effects of MOS regulating rumen fermentation.

Hematological parameters are typically auxiliary indexes used to monitor health conditions or metabolic processes of animals. When used as an animal feed additive, supplementation dietary MOS improves digestion, the immune system, and the microbiota of the gastrointestinal tract. However, because their effects on the animal body are slight, basal hematological parameters are not greatly influenced by MOS. In the present study, supplementation dietary MOS did not influence the hematological parameters of sheep (refer to Table 6). Similarly, supplementation dietary MOS did not affect the WBC count, HCT, neutrophil count, mononuclear leukocyte count, or eosinophil count of dairy cows or their offspring (Franklin et al., 2005). Other studies on the effects of MOS in monogastric animals have reported similar results: supplementation dietary MOS did not impact RBC count, hemoglobin, HCT, or MCHC of weanling pigs in one study (Valpotić et al., 2017; Dos Anjos et al., 2019), or RBC count, WBC count, lymphocyte count, or diarrhea score in another study (Zhao, Jung & Kim, 2012), and did not influence monocyte, basophil, or eosinophil counts in Arbor Acres broiler chickens (Attia et al., 2017).

Conclusions

In the current study, the efficiency of supplementation dietary MOS on growth performance, nutrients apparent digestibility, rumen fermentation, and some hematological parameters were assessed in Hu sheep under actual breeding conditions. Although supplementation dietary MOS did not affect growth performance, ruminal SCFAs concentration and ratios of individual fatty acids to total SCFAs, C2/C3 ratios, or hematological parameters in sheep, these oligosaccharides increased DM, OM, CP, NDF, ADF, and ash apparent digestibility and decreased the ruminal ammonia concentration significantly. This indicates that supplementation dietary MOS improve nutrient utilization, especially nitrogen metabolism, in ruminants. In the future, the ruminal microbiota, ruminal microorganism metabolism, and relationship between gastrointestinal microorganism and host regulating by MOS should be investigated, and eventually reveal the molecule mechanism.

Supplemental Information

Supplemental Information 1 Growth performance of sheep.

Click here for additional data file.

Supplemental Information 2 Nutrients apparent digestibility and retention rate of sheep.

Click here for additional data file.

Supplemental Information 3 Ruminal short-chain fatty acids of sheep.

Click here for additional data file.

Supplemental Information 4 Ruminal ammonia concentration of sheep.

Click here for additional data file.

Supplemental Information 5 ARRIVE 2.0 checklist.

Click here for additional data file.

We thank everyone who participated in this experiment. We would also like to thank Editage for English language editing.

Additional Information and Declarations

Competing Interests

Author Contributions

Animal Ethics

Data Availability

The authors declare that they have no competing interests.

Chen Zheng conceived and designed the experiments, performed the experiments, analyzed the data, prepared figures and/or tables, authored or reviewed drafts of the paper, and approved the final draft.

Juwang Zhou performed the experiments, analyzed the data, prepared figures and/or tables, and approved the final draft.

Yanqin Zeng performed the experiments, analyzed the data, prepared figures and/or tables, and approved the final draft.

Ting Liu conceived and designed the experiments, performed the experiments, analyzed the data, prepared figures and/or tables, authored or reviewed drafts of the paper, and approved the final draft.

The following information was supplied relating to ethical approvals (i.e., approving body and any reference numbers):

All experimental protocols and sample collection were approved by the Ethics Committee of Gansu Agriculture University under permission no. GAU-LC-2020-018.

The following information was supplied regarding data availability:

The raw data are available in the Supplemental Files.

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
