# Peer review of "Effects of mannan oligosaccharides on growth performance, nutrient digestibility, ruminal fermentation and hematological parameters in sheep"

_PeerJ, doi:10.7717/peerj.11631_

## Round 0.1 · original submission · Major Revisions

Authors, kindly address the comments raised by the reviewers.
Importantly, the editor raises serious concerns about the contribution of this work, and therefore requests authors to clarify as well as justify the difference between this manuscript, and a previously published work at: J Animal Sci 2018, 96: 284-292 (https://dx.doi.org/10.1093/jas/skx040), which reports a very closely related topic. This must be clarified in great detail in your response, as well as at the introduction of this work to justify the rationale for this study before the objective statement.
Thank you

Reviewer 1 ·

Basic reporting

Although the manuscript highlighted the importance of feed supplementation with prebiotics like Mannan Oligosaccharide (MOS) in sheep diet which improves nutrient utilization and decreases ammonia-nitrogen production in the rumen. The following observations were made:
1.The article should include sufficient introduction and background to demonstrate how the work fits into the broader field of knowledge because the majority of the previous work cited about the role of MOS are related to monogastric and poultry species which have different anatomy and physiology with ruminants (Sheep). It should also state clearly the statement of the problem, hypothesis, and justification for conducting such research. Therefore, relevant prior literature should be appropriately referenced.

Experimental design

1. The methodology adopted needs to be reference from previous studies for each procedure either with modification or not because it wasn’t mentioned in the manuscript.
2. In line161-162 of the manuscript, the side of blood collection used (Cervical vein) is not appropriate for blood collection in sheep rather (jugular vein). Additionally, blood collected in non-heparinized tubes will be clotted leaving only serum and not suitable for hematological examination.

Validity of the findings

1. The results presented were Satisfactory.

2. There is an incorrect heading for Table 3. Where serum was harvested from blood but hematocrit parameters were presented. Therefore, no result for serum biochemistry presented.

Additional comments

I suggested to the author to improve on defining his research question and proving his hypothesis and all references of previous work given should be consistent with his research objectives.

Reviewer 2 ·

Basic reporting

No comment.

Experimental design

No comment.

Validity of the findings

No comment.

Additional comments

i) I recommend the authors to further clarify how MOS increased nutrient digestibility.

ii) In line 226, citation or reference is needed for the statement "MOS are a type of carbohydrate that can be degraded by ruminal microbes."

iii) Generally, elaborate discussion of the results is recommended.

Reviewer 3 ·

Basic reporting

The manuscript is well prepared and the tables are nicely configured. Authors just need to work on the decimal places in the tables to make it neater. Literature are well cited and adequate background information provided

Experimental design

The design is ok
Just a little concern with the statistical analysis

Validity of the findings

no comment

Additional comments

Review of manuscript 'effects of manna oligosaccharides on growth performance, nutrient digestibility, ruminal fermentation and hematological parameters in sheep.
Generally, the study is well laid out, properly designed with adequate number of animals. There are just a few issues I will like to point out, for example in the statistical analysis, the authors can use the mixed model approach to analyse the data because the ingredients in the formulated diet are varied. Also there are improvement in the DM, OM, Ash etc apparent digestibility of animals fed the MOS treatments but no effect on the average daily gain. Mixed Procedure of SAS will be able to account for some of the errors.
Table 1
Looking at the ingredient table, normally animals on 1.6 and 2.4% MOS will have better performance taking account the feed intake which will be more palatable considering extra 50kg maize, 30 kg soybean, lesser maize straw and malt sprout. Although authors tried to balance in terms of chemical composition, but there is a little concern with the feed formulation. Authors could have included a treatment as positive control with a commercial supplement vs MOS, while presenting the ingredient composition
The chemical composition g/kg should be presented in terms of DM composition. The DM should be removed and other nutrient presented in terms of the DM
Why was acid detergent fibre not determined?
Table 2
Average intake or average daily intake?
The unit should of average daily intake should be in g/d
Initial weight is missing in this table

Table 3
The title is not properly written and carried no information about the treatments
Values in the table should be standardized, 1117 instead of 1117.94, and 367.9 instead of 367.8, in that way at least there will be 4 numbers and the table becomes neater

Line 120, How was the doses determined?
Line 154, how did you ensure rumen fluid collected were not contaminated with saliva? And How were you able to take rumen fluid samples from 3 regions of the rumen when using stomach tube?

Experimental diets
The authors claim in line 120 that the experiment was a single-factor design, meaning the only introduction was the 4 different doses of MOS, whereas in line 130, authors formulated 4 diets and varied the composition of ingredients in the diet. The different proportion of ingredient composition could have one way or the other influence response.

Line 135 to 139, Authors carried out digestion and metabolism trial during the short 5 weeks study. Authors could have carried out the digestion and metabolism trial after the growth performance trial as the break for the digestion trial could have influenced the experimental animals

Line 144 to 145, How did the authors determine the DM of the faecal samples

Blood sample analysis, authors left out serum biochemical parameters, parameters such as blood urea nitrogen and liver enzymes such as ALT, ASP could give an indication of the health status of the animals

Line 187, delete “The ash apparent digestibility of sheep fed the 0.8% MOS diet was higher than those in sheep fed the 2.4% MOS diet (P = 0.034)”
Line 196, “MOS did not affect the physiological or biochemical parameters of sheep serum”, authors did not test the serum biochemical indices of sheep
Line 201 to 209, unusually long sentence, too long, try and breakdown into shorter sentences
Line 219 to 222, “This may be because ruminal microbes degrade MOS in the rumen, preventing it from affecting the rest of the body (although it can still improve nutrient utilization and ruminal fermentation). This indicates that MOS are suitable for nutritional regulation purposes in adult ruminants, but not for growth performance enhancement”
This is self-contradictory, authors need to rephrase
Line 225 to 228, I don’t really understand what the authors meant by the statement. Of course most carbohydrates can be degraded in the rumen, this is useful for the microbes to multiply and make their cells, this also leads to production of volatile fatty acids which contributes mainly to the energy requirements of the animal, and for milk production. Authors need to bring clarity to this.

---

## Round 0.2 · Minor Revisions

Reviewers have assessed your work, and have recommended further corrections. Please kindly attend to them. Looking forward to your revised manuscript.

Reviewer 1 ·

Basic reporting

The manuscript now looked scientifically good and interesting as the introductory part identified the research problem and clearly stated the hypothesis in question. I believe this manuscript merits publication but I have only a few comments and suggestions for the authors

Experimental design

No any comments

Validity of the findings

Satisfactory

Additional comments

In the last review, I suggested these observations to the author but still, they appeared again in the second review, therefore, 1. The authors should give clarification of mentioning four doses instead of three (3) and control because for the first dose no MOS was added rather a MOS-free diet.

2. The author should have mentioned twenty-four (24) animals per group instead of 6 replicate with 4 replicate each.

3. The authors should give the scientific basis for using non-heparinized vacuum tubes in collecting blood and how he measured hematological indices with clotted blood.

4. The author should differentiate between the terms he used in the abstract hematological parameters, in methodology blood parameter, and in result physiological and biochemical parameters as to whether they mean the same or different meaning?

Reviewer 3 ·

Basic reporting

no comment

Experimental design

no comment

Validity of the findings

the validity is ok

Additional comments

I have read through the manuscript again. Although the manuscript has been improved from the original state, I still have some concerns I raised before, especially with the feed formulation. Formulation of feed for ruminants is not only about the chemical composition but also the acceptability. It became obvious that there will be a confounding effect. I can judge this without even conducting the experiment based on more maize, soybean, lesser barley, and wheat straw. Six ingredients have been varied out of 12 ingredients used.
Secondly, I think there is a need to send the response letter to English language editor as I struggle keeping up with some of the responses.
Thirdly, the authors need to motivate in the manuscript why carrying out digestibility study during a 5 week trial is good, preferably linking it with a reference

---

## Round 0.3 · Minor Revisions

Reviewer’s response to your revision is positive towards publication. However, the editor would like authors to address few other concerns to help elevate the quality of this work further.

a) The Introduction needs further strengthening. Information regarding growth performance, nutrient digestibility, ruminal fermentation, and hematological parameters in sheep is not sufficient. Kindly provide SUCCINCT literature synthesis about each of these parameters, what it means in sheep’s wellbeing , each must clearly stand out, so that readers will understand why these parameters are being tested in in the materials and methods, and why there are the point of discussion. Apply your discretion on how to arrange it. The editor will look out for these in your introduction.

b) The reason ‘why’ this study is relevant is not so strong, improve the justification of this study. Why is this study being conducted? Make it to stand out clearly. This must be before the objective statement and both have to be together (and alone) as the last paragraph of the introduction.

c) The Editor, having carefully read your rebuttal note, it is important that your materials and methods need to be improved to help readers understand your work better. Start the materials and methods with a subsection captioned ‘Schematic overview of the experimental program’ . This subsection should comprise 3-4 sentences and must be supported by a schematic /flow diagram which will be Figure 1 . The idea here is to provide a snapshot from how this samples were collected, , prepared, allocated to analysis . You have four different doses, provide this diagrammatically to show each 24 samples. The flow diagram must show the test period, and at what points were analysis carried out. Apply your discretion on how the diagram should be. Remember to succinctly describe this diagram and it must directly connect with the objective of this study.

d) Line 138: sample collection and analysis as a title of this subsection is not appropriate. Authors please do this for lines 139-174. Break up this section into its respective sub-subsections. How the samples were prepared, should all be under the sub subsection captioned ‘sample preparation’. All the analyses, and which analyses did you perform, each of them must stand out with each having its own caption, and its description, and all must be under ‘analytical methods’. This will bring better organization to this section. The editor will look out for this in your revised manuscript.

e) In the statistical analysis, it will help readers if you can provide a sentence or two why this ANOVA model was used. Not all readers are experts in this specific scientific endeavor. Some would most likely be learners. It will be wise to differentiate probability value (P value) and tendency value (P with smaller ‘t’ value) throughout the text so that readers can clearly differentiate them.

f) Indicate where you would like Tables and Figure to be placed using ‘(Please place Table 1 here)’ ‘(Please place Figure 1 here)’

g) In the discussion, kindly indicate which specific Tables are being referred to. Use ‘ (Refer to Table?)’ where specific result is being mentioned in the discussion. This is to specifically guide readers to better and effectively connect with specific areas of the results being discussed.

h) In the conclusion , please start it by reiterating why this study was conducted, and its objective, before line 292 begins. Authors are encouraged to suggest what the direction of future study / studies ought to look for?

This is a very good study. With the above advice, the quality of the work will further improve. Authors are encouraged to carefully address all in its required detail. Look forward to your revised manuscript.

Thank you very much.

Reviewer 3 ·

Basic reporting

Satisfactory

Experimental design

The study and its experimental design falls within the aims and scope of the journal, the hypothesis are well defined and the objective clearly indicated. Standard methods have been used to carry out the experiment

Validity of the findings

Satisfactory

Additional comments

The authors responses to review suggestions and criticism are satisfactory

---

## Round 0.4 · accepted · Accept

Thank you for revising your work, and addressing all concerns. Authors have benefitted very well from the peer-review process. It is now acceptable for publication. Thank you for finding PeerJ as your journal of choice, and look forward to your future submissions.
Congratulations